# Microbiome—Friend or Foe of Pancreatic Cancer?

**DOI:** 10.3390/jcm10235624

**Published:** 2021-11-29

**Authors:** Jaroslaw Daniluk, Urszula Daniluk, Pawel Rogalski, Andrzej Dabrowski, Agnieszka Swidnicka-Siergiejko

**Affiliations:** 1Department of Gastroenterology and Internal Medicine, Medical University of Bialystok, 15-276 Bialystok, Poland; pawel.rogalski@umb.edu.pl (P.R.); adabrows@umb.edu.pl (A.D.); agnieszka.swidnicka-siergiejko@umb.edu.pl (A.S.-S.); 2Department of Pediatrics, Gastroenterology, Hepatology, Nutrition and Allergology, Medical University of Bialystok, 15-274 Bialystok, Poland; urszula.daniluk@umb.edu.pl

**Keywords:** pancreatic cancer, microbiome, inflammation, immunotherapy

## Abstract

Pancreatic ductal adenocarcinoma is one of the deadliest human neoplasms. Despite the development of new surgical and adjuvant therapies, the prognosis remains very poor, with the overall survival rate not exceeding 9%. There is now increasing evidence that the human microbiome, which is involved in many physiological functions, including the regulation of metabolic processes and the modulation of the immune system, is possibly linked to pancreatic oncogenesis. However, the exact mechanisms of action are poorly understood. Our review summarizes the current understanding of how the microbiome affects pancreatic cancer development and progression. We discuss potential pathways of microbe translocation to the pancreas, as well as the mechanism of their action. We describe the role of the microbiome as a potential marker of pancreatic cancer diagnosis, progression, and survival. Finally, we discuss the possibilities of modifying the microbiome to improve treatment effectiveness for this deadly disease.

## 1. Introduction

Due to the development of modern molecular techniques in recent years, we are rediscovering the world of microorganisms, including microbiota. The human microbiota is defined as the group of microbes that colonize various sites of the human body and its ecosystem. The term microbiome refers to genetic information encoded by microbiota, its ecosystem, and the host environment; however, many scientists use the words microbiota and microbiome interchangeably. Among several niches, the digestive tract contains the richest community of microbiota exceeding 10^14^ microorganisms [1]. Most of them are harmless and involved in many physiological processes such as the digestion of nutrients, production of vitamins and other high-energy substrates, modulation of the immune system, and/or protection against pathogenic bacteria [2]. However, any disturbances in microbiota composition, also known as dysbiosis, are associated with numerous diseases such as obesity, diabetes, and inflammatory bowel disease. Moreover, the participation of dysbiosis in cancers, such as laryngeal, gastric, colorectal, and liver cancer, has recently been emphasized [3,4]. It is assumed that 16% of all human malignancies are attributed to infectious agents, which are an integral component of microbiota [5]. There is now emerging evidence that dysbiosis is also very closely related to pancreatic oncogenesis.

Pancreatic ductal adenocarcinoma (PDAC) is the 12th most common and one of the deadliest neoplasms in humans. It is estimated that by 2030 PDAC, will be the second leading cause of cancer death in the United States [6]. Despite the constant progress in treatment, the overall survival rate of this tumor is 9% [7]. Only 20% of pancreatic cancer patients are suitable for surgery. Unfortunately, even in such cases, the 5-year survival does not exceed 25% [8]. The reason for this very poor outcome is the lack of early clinical symptoms, metastatic spread, resistance to chemotherapy, as well as a high recurrence rate after surgery. Moreover, PDAC has a very specific tumor microenvironment with a dense extracellular matrix, which consists mainly of fibroblast and protumoral immune cells [9]. Because cases of pancreatic cancer, attributable to known preventable risk factors, such as obesity, diabetes, chronic pancreatitis, or genetic disorders, have been shown to range from 24% of cases in China to 36% in the United Kingdom, other causes of PDAC are being urgently investigated [10,11]. Recent studies link pancreatic tumor development to disturbances in human microbiota [12,13]. Several possible interactions between dysbiosis and pancreatic oncogenesis have been proposed; however, the exact mechanisms are still unknown. In this review, we summarize the recent data from human and animal studies describing the pancreatic microbiota in health and disease, potential mechanisms of microbial colonization in the pancreas, and microbial contribution to PDAC development. In addition, we discuss the possible usefulness of microbiota as a diagnostic and predictive biomarker, as well as the implications of microbiota modification in the treatment of this deadly disease.

## 2. Does the Pancreas Have Its Own Microbiota?

For a long time, it has been assumed that microorganisms could not survive in the pancreatic juice, due to its alkaline pH and numerous digestive enzymes. However, this paradigm has been recently changed. Several research groups have analyzed the microbiome of pancreata obtained during surgery or from organ donors [12,13,14,15]. Pushalkar et al. found a 1000-fold increased bacteria abundance in both mice and human PDAC compared to a normal pancreas [13]. The bacterial composition and diversity of normal and cancerous pancreata were also different. Thirteen distinct phyla were present specifically in human PDAC, mostly *Proteobacteria*, *Bacteroidetes*, and *Firmicutes* [13]. Moreover, the composition of a pancreatic microbiome at an early stage of the disease was distinct from that of an advanced PDAC, suggesting that cancer progression is linked to changes in the microbiome. Using both taxonomic profiling via 16S rRNA gene sequencing and fluorescence in situ hybridization (FISH) techniques, Riquelme et al. found a significant abundance of intratumoral bacteria in surgically resected PDAC specimens, with enrichment of *Proteobacteria* and *Actinobacteria* at the genus level [12]. They also compared the microbiome from gut, tumor, and normal adjacent tissue in these patients and showed that the human gut microbiome represents 25% of the tumor microbiome, but not of the normal tissue. Based on their results, high microbial diversity may affect the natural course of the disease. Geller et al. detected pancreatic microbiome significantly more often in the cancerous pancreas than in normal tissues from organ donors (76% vs. 15%, *p* < 0.005) and more than half of them belonged to the class *Gammaproteobacteria* [16]. The presence of the microbiome in normal and cancerous human pancreases was confirmed in another study; however, in this case, there was no difference in species richness or beta diversity between the normal and the tumorous tissue [15]. A comprehensive analysis of seven common cancers found PDAC to have the highest microbiome presence (68.2%) of all of them [17]. Importantly, these bacteria were located mostly intracellularly in both cancer and immune cells. Moreover, the presence of the microbiome was also confirmed in benign pancreatic lesions that can progress to cancer. Analysis of cystic fluid from a surgically resected pancreatic cystic neoplasm revealed a 10-fold increase in bacterial DNA copies in intraductal papillary mucinous neoplasms (IPMNs) compared to non-IPMNs lesions [18]. The abundance of bacteria was positively correlated with the degree of dysplasia within IPMNs.

However, the microbiome is not just bacteria. It also includes viruses, fungi, and archaea. In a Taiwan population-based cohort study, patients with candidiasis had a significantly increased risk of pancreatic cancer compared to individuals without *Candida* infection (adjusted hazard ratio = 2.39, 95% confidence interval, CI = 1.09–5.24) [19]. Another prospective cohort study from Sweden revealed that patients with *Candida*-related oral lesions had 70% excess risk of developing pancreatic cancer compared to healthy controls [20]. It has been recently reported that cancerous pancreases of mice and humans possess 3000-fold increased fungi colonization compared to a normal pancreas [14]. Moreover, the mycobiome composition, based on alpha- and beta-diversity, was different between malignant and normal pancreatic tissue. In particular, the abundance of *Malassezia* species was much higher in pancreatic cancer, and the repopulation of mycobiome ablated *Kras* mice with *Malassezia* species, but not with *Candida*, *Aspergillus*, or *Saccharomyces* species, significantly accelerated PDAC progression [14].

Viruses are another component of the microbiota that may have a significant impact on the process of pancreatic oncogenesis. A recent study by the Pan-Cancer Analysis of Whole Genomes Consortium, examining genome-wide sequencing data from 38 types of cancer, found a high prevalence of known tumor-associated viruses [21]. In pancreatic cancer in particular, the most common viruses detected in the tumor were *Roseoloviruses*, *Lymphocryptoviruses*, and *Gammaretroviruses*. Other components of the viral microbiome that may be involved in the development of PDAC are the hepatitis B virus (HBV) and the hepatitis C virus (HCV), well-known oncogenic viruses. Because the pancreas and liver share many blood vessels and a common duct outlet into the duodenum, the pancreas is a potential target of hepatotropic viruses. Indeed, clinical studies confirmed the presence of both the hepatitis B surface antigen (HbsAg), a marker of HBV infection and HCV antigen in pancreatic acinar cells [22]. Several studies assessing a link between viral hepatitis and pancreatic cancer have been conducted, but they provided inconclusive results [23,24]. However, the meta-analysis of eight case-control and two cohort studies, showed a relationship between HBV or HCV infection and an increased risk of PDAC [25]. Undoubtedly, we need more clinical data to assess the relationship between viral infection and pancreatic oncogenesis.

Taken together, the available data from experimental and clinical studies clearly indicate the pancreas harbors its own microbiota. However, we need more research to resolve this chicken and egg paradox. It should be investigated whether the development of PDAC is caused by an abnormal microbiota (which in our opinion is more likely), or whether pancreatic cancer and immunological abnormalities allow the colonization of the pancreas by pathogenic bacteria.

## 3. How Does Microbiota Colonize the Pancreas?

There are three possible methods of microbiota transmission into the pancreas. The pancreas naturally connects to the digestive tract through the Wirsung duct and communicates with the liver through the common bile duct. Microorganisms of the upper gastrointestinal tract may gain access to pancreatic parenchyma via the pancreatic duct. Pushalkar et al. administered to mice, via oral gavage, a fluorescently labelled *Enterococcus faecalis*, and found their translocation into the pancreas as early as 30 min later [13]. Similarly, Aykut et al. demonstrated that the fluorescently tagged fungal strain *Saccharomyces cerevisiae*, delivered to the gut via oral gavage, was detected in the pancreas of wild-type and *Kras* mice within 30 min [14]. On the contrary, in wild-type germ-free mice transferred to specific pathogen-free conditions, the presence of bacteria within the pancreas was not observed despite bacterial oral gavage for 8 weeks [15]. A possible explanation for these conflicting results is that the latter study used mice that lacked the mutations in the pancreas responsible for the development of PDAC, which may be a factor in favor of the microbiome colonization of the pancreas. It can be speculated that the development of pancreatic pathologies related to certain mutations, both pancreatitis and cancer, combined with increased bacterial translocation are key factors for pancreatic microbiome colonization.

The hypothesis of bacterial translocation through the papilla of Vateri was supported by a clinical study [18]. Bacterial DNA was found in patients with IPMNs, which are the lesions that communicate with the pancreatic ductal system, but not in other cystic lesions that do not drain to the pancreatic duct. Moreover, further analysis demonstrated the presence of oral bacterial taxa (*Fusobacterium nucleatum* and *Granulicatella adiacens*) within IPMN cystic fluid, which supports the theory of bacterial translocation from the upper GI tract via the pancreatic duct. Undoubtedly, additional tests are necessary to confirm or exclude the possibility of microbiota transmission through the pancreatic duct. An alternative route of pancreatic microbiota origin may be the lower gastrointestinal tract. Microorganisms can translocate through the portal circulation or mesenteric lymph nodes. One of the clinical pieces of evidence to confirm this theory is the fact that in necrotizing acute pancreatitis, organ infection is mainly caused by bacteria that are characteristic for the lower digestive tract [26].

So far, how the pancreas is colonized by microbiota remains unclear. Understanding these mechanisms may have important clinical implications. If microorganisms enter the pancreas through the papilla of Vateri, it may be beneficial to use antibiotic prophylaxis (systemic or local) in patients undergoing endoscopic procedures in the biliary/pancreatic ducts to prevent bacterial colonization of the pancreas. Undoubtedly, we need more clinical and experimental data to resolve this issue.

## 4. How Does the Microbiome Influences the Carcinogenesis Process in Pancreas?

To determine the role of the microbiome in cancer progression, an animal model of mice with pancreatic mutant *Kras* expression was used [13]. These animals spontaneously develop pancreatic cancer. However, germ-free mutant *Kras* mice, in other words animals without a microbiome, showed delayed pancreatic dysplasia. Moreover, tumor progression was significantly accelerated when the bacterial gut of *Kras* mice was ablated with antibiotics, and subsequently repopulated with feces derived from mice expressing both mutant intrapancreatic *Kras* and *Trp53* genes, but not from wild-type mice [13]. Similarly, in an animal model of mice with concomitant *Kras* mutation and partial loss of *PTEN* tumor suppression (Kras^G12D^/PTEN^−/+^), bacterial ablation with antibiotics resulted in a decreased rate of PDAC development [15]. To assess whether the progression of PDAC is influenced by the presence of pancreatic or intestinal microbiota, an orthotopic model of a pancreatic tumor was used. In wild-type mice (WT), oral antibiotics treatment resulted in a 50% reduction of orthotopic pancreatic tumor size [13]. This finding suggests that intestinal microbes have the ability to promote pancreatic development via systemic action.

The exact mechanisms by which the microbiome influences the development of pancreatic cancer have not been discovered yet, but several theories are under consideration. In one hypothesis, deoxycholic acid, a secondary bile acid produced by intestinal bacteria, induces DNA damage and leads to pancreatic cancer development [27]. Another theory is that microbiota may promote carcinogenesis by inducing chronic inflammation. PDAC is considered to be an inflammation-driven neoplasm. Although the *Kras* mutation occurs in nearly all pancreatic cancers, it is insufficient for tumor initiation on its own. Using an animal model of mice with pancreatic *Kras* mutations, we have shown that inflammatory stimuli, lipopolysaccharides (LPS), clinically relevant inflammatory inducers associated with Gram-negative bacteria, initiate a positive feedback loop involving NF-κ B that amplifies Ras activity and promotes carcinogenesis [28]. These data suggest that the microbiome enhances oncogenic signaling by promoting pancreatic inflammation. This theory may explain the mysterious link between disturbances in the composition of the oral microbiota leading to periodontal disease (PD) and an increased risk of PDAC [29,30]. Meta-analysis of eight studies from Europe, Asia, and North America revealed a 74% increased risk of PDAC in individuals with periodontitis and a 54% higher risk of cancer in patients with edentulism even after adjusting to potential confounding factors such as age, sex, tobacco smoking, alcohol, diabetes, and BMI [31]. Similarly, in a prospective study of male health professionals, PD was associated with a 64% increased PDAC risk [32]. *Porphyromonas gingivalis* (*P*. *gingivalis*), a Gram-negative bacterium implicated in the pathogenesis of human periodontitis, can activate toll-like receptors 4 (TLR4) to enhance the release of proinflammatory cytokines, such as interleukin 1β, interferon γ, and the tumor necrosis factor, to promote pancreatic inflammation and, subsequently, pancreatic cancer [33,34]. However, it cannot be ruled out that the oral microbiota promotes carcinogenesis through circulating oral microbial toxins [35]. *P. gingivalis* produces an enzyme that degrades arginine and promotes *p53* and *Kras* mutations [36]. The exact mechanisms leading to an increased risk of PDAC by oral pathogens require further investigation.

The mechanisms responsible for fungal-induced oncogenesis are also complex. Mannose-binding lectin (MBL) is a pattern recognition receptor of the innate immune system that recognizes fungal pathogens. Its binding to glycans of the fungal wall activates a lectin pathway and a complement cascade that triggers C3 convertase [37]. Aykut et al. found that mycobiota promotes pancreatic oncogenesis via activation of the MBL–C3 axis, and the deletion of MBL or C3 in the extratumoral compartment were both protective against tumor growth [14]. Similarly, MBL- and C3-deficient mice were also protected against PDAC progression. Another fungal representative, *Candida*, can promote PDAC development through the production of well-known carcinogens, such as nitrosamines, the induction of an inflammatory response, or the activation of the Th17 response and molecular mimicry [19].

Moreover, increasing evidence suggests a crucial role of the microbiome in modulating the immune landscape of the tumor microenvironment, which is highly immunosuppressive [15,38]. Microbiome-induced chronic inflammation suppresses the innate and adaptive immune systems, through the activation of Toll-like receptors (TLRs) and the increase in tumor-promoting myeloid-derived suppressor cells (MDSC) and M2 macrophages [13]. The consequence of this phenomenon is an excessive differentiation of suppressive populations of CD4+ T cells and a decreased number of cytotoxic CD8+ T cells, which ultimately promotes cancer progression. In an animal model of orthotopic pancreatic cancer, gut microbiome eradication by antibiotics reduced subcutaneous tumor growth and liver metastases, which was related to a significant increase in interferon gamma-producing T cells and a decrease in IL-17A and IL-10 producing T cells [39]. Riquelme et al. determined that the crosstalk between the tumor and the intestinal microbiome shapes the immune tumor microenvironment and immune responses [12]. They found a significant correlation between a higher number of CD3+ and CD8+ T cells, as well as granzymeB+ cells and the improved overall survival in PDAC patients. Moreover, tumor microbiome diversity and the presence of *Saccharopolyspora*, *Pseudoxanthomonas*, and *Streptomyces* within tumor tissue were both positively corelated to the anti-tumor immune response via the recruitment and activation of CD8+ T cells. In an animal model of orthotopic pancreatic cancer, mice that received fecal microbiota transplantation from long-term-survival patients had significantly higher numbers of CD8+ T cells, activated T cells (CD8+ IFNg+ T cells) within the tumor, as well as higher serum levels of interferon-γ and IL-2 compared to mice that received fecal microbiota from short-term-survival patients [12].

Taken together, several studies emphasized a potential link between the gut or oral microbiota and pancreatic oncogenesis; however, these data need to be interpreted with caution. Since most of the research on this topic has been conducted on mice, we cannot directly extrapolate these results to humans. We need large-cohort clinical studies to determine the effect of geographical, ethnic, and environmental (i.e., diet, smoking, antibiotics) variances on microbiome-PDAC crosstalk. Determining whether microbiota promotes oncogenesis through direct mutagenic/toxic effects or through immunomodulation may have important therapeutic implications.

## 5. Microbiota as a Potential Biomarker of Pancreatic Cancer

The detection of premalignant lesions or early stages of pancreatic cancer may improve survival; however, available biomarkers have very limited sensitivity and specificity, and because of that they cannot be used for screening purposes. There is growing interest in applying changes of the microbiota to distinguish healthy individuals from patients with PDAC. Differences in oral, gut, pancreatic, or blood microbiota composition and diversity in patients with early or advanced pancreatic cancer as well as with precancerous lesions are of great importance (Table 1).

Molecular analyses of samples from The Cancer Genome Atlas for 33 types of cancer found unique signatures of microbes in the blood within and between most major cancer types [46]. In melanoma, prostate cancer, and lung cancer, solely using plasma-derived, cell-free microbial nucleic acids, the authors could discriminate among cancer patients and healthy individuals. Extracellular vesicles (EVs) are produced by the microbiota and contain a large amount of bacteria-derived genomes. Due to their very small size, EVs can travel from the intestinal cellular membrane to the bloodstream [47]. To develop a new microbial biomarker for the early detection of PDAC, Kim et al. performed a 16S rRNA gene analysis and compared microbial EVs acquired from blood samples from pancreatic cancer patients and healthy controls [45]. Significant differences in microbiota composition at the phylum level were demonstrated, with an increase in *Verrucomicrobia*, *Deferribacteres*, *Bacteroidetes*, and a decrease in *Actinobacteria* in PDAC patients. Applying the combination of the microbiome markers, a PDAC prediction model was developed with a high area-under-the-receiver-operating-characteristic curve (0.966 and 1.000, at the phylum and genus levels, respectively) [45].

Another promising method for the early detection of PDAC is the analysis of pathogenic bacteria responsible for the development of periodontitis (antibodies in the blood or oral microbiome), which as mentioned, may serve as predictors of pancreatic cancer development. In a large European prospective cohort study, Michaud et al. measured antibodies to 25 oral bacteria in plasma samples collected from patients prior to pancreatic cancer onset (the mean time between blood draw and the date of diagnosis was 5.0 years) and compared them to healthy controls [29]. Individuals with high levels (>200 ng/mL) of antibodies against *P*. *gingivalis* (ATTC 53978 strain) had a twofold higher risk of pancreatic cancer than individuals with lower levels (≤200 ng/mL) of these antibodies. To explore the association with commensal (non-pathogenic) oral bacteria, the authors performed a cluster analysis and identified two groups of individuals, based on their antibody profiles. A cluster with higher overall levels of antibodies to commensal oral bacteria had a 45% lower risk of pancreatic cancer than the cluster with lower overall levels of antibodies. Oral wash samples were used to determine the composition of the oral microbiome (using 16S rRNA gene sequencing) in individuals from two large prospective cohort studies [30]. The analysis of 361 PDAC cases collected prior to the disease onset and 371 matched controls revealed that the presence of *P*. *gingivalis* (OR 1.60; 95% CI 1.15–2.22) and *Aggregatibacter actinomycetemcomitans* (OR 2.20; 95% CI 1.16–4.18) were correlated with an increased risk of PDAC development. Moreover, the phylum *Fusobacteria* and its genus *Leptotrichia* decreased the risk of pancreatic cancer. *Neisseria elongata* (*N*. *elongata*) and *Streptococcus mitis* (*S*. *mitis*) are the other components of the oral microbiome that could potentially be useful in the diagnosis of pancreatic cancer. Farrell et al. performed salivary microbiome profiling and found that the abundance of *N*. *elongata* and *S*. *mitis* were significantly decreased in patients with PDAC compared to healthy subjects [40]. The combination of these two bacterial strains resulted in 96.4% sensitivity and 82.1% specificity in distinguishing patients with pancreatic cancer from healthy individuals. Understanding which bacteria are related to the risk of pancreatic cancer may help improve early disease detection.

Previous studies that analyzed the pancreatic microbiome in PDAC carry a potential risk of bias as they are limited to the samples obtained during surgical resection. As only one in five patients with pancreatic cancer is eligible for surgery, the results of such studies are not representative of the entire group of patients with PDAC. Therefore, a new proof-of-concept study was performed to determine the feasibility of using pancreatic formalin-fixed paraffin-embedded (FFPE) samples obtained during an endoscopic ultrasound fine-needle biopsy (EUS-FNB) to estimate the pancreatic microbiome [48]. In this pilot study, 17 EUS-FNB samples (9 samples with PDAC) were analyzed. The authors used Decontam, a bioinformatics tool, to identify and remove contaminant DNA sequences, and were therefore able to identify 275 operational taxonomic units. Unfortunately, no differences in alpha and beta diversity were found between PDAC patients and the control group, but this method is undoubtedly in a very promising direction for obtaining material for pancreatic microbiome analysis. Moreover, EUS-guided fine-needle injection is a potential route of drug delivery into the tumor.

An alternative source of new pancreatic cancer biomarkers is the gut microbiome profile. A distinct intestinal microbiome composition has been found in animal models of PDAC [13,14,15]. In a prospective clinical study, differences in the gut microbial composition between PDAC patients and healthy controls in a Chinese cohort were determined by MiSeq sequencing [43]. PDAC individuals had a unique gut microbial profile and decreased alpha diversity. Based on beta diversity, pancreatic cancer specimens were enriched with certain pathogenic and lipopolysaccharides-producing bacteria with a concomitant decrease in probiotics and butyrate-producing bacteria. Moreover, the combination of 40 increased PDAC genera had a very high classification capacity with an AUC of 0.842 between pancreatic cancer and healthy controls [43]. In another study, the fecal microbiota of 30 patients with PDAC, 6 patients with pre-cancerous lesions, 16 individuals with non-alcoholic fatty liver disease, and 13 healthy subjects was analyzed in an Israeli cohort [44]. Fourteen bacterial features discriminated between PDAC and controls, with partial agreement with the previously described Chinese cohort study [43]. The Random Forest algorithm based on the microbiota classified PC and control samples with an AUC of 0.825 [44]. However, large interpersonal variation in the composition of the microbiome was observed, with a small number of bacteria common to both neoplastic and precancerous lesions.

Currently, there is no effective screening strategy for pancreatic cancer. In most cases, the available diagnostic methods do not allow for the detection of the disease at a curable stage. Therefore, the possibility of using the microbiota as a potential biomarker seems to be a very promising alternative. However, there are still many challenges to overcome. One of the main problems is the diversity of the composition of the microbiota between studies, which does not allow for establishing uniform tests for patients with different demographic and geographical characteristics. Therefore, a predictable oral, gut, or pancreatic microbiome in pancreatic cancer does not yet exist. In addition, a predictable microbiome must distinguish PDAC not only from healthy individuals but also from other chronic diseases. Therefore, further large, prospective, multi-center studies are needed to fill our knowledge gap about the specific PDAC microbiome.

## 6. Tumor Microbiota as a Predictor of Survival

Niels Bohr, the Nobel laureate in Physics, stated that “Prediction is very difficult, especially if it’s about the future”. Unfortunately, it is much easier to predict survival in pancreatic cancer, as 91% of all patients survive less than 5 years and surgical resection modestly improves the outcome [7]. On the other hand, it remains unclear what factors influence the long-term survival in a minority of PDAC patients. Current data suggest that the pancreatic microbiome may be related to the prognosis of PDAC. RNA-sequencing data from 187 PDAC patients revealed that 13 bacterial species, mainly from the phylum *Proteobacteria*, were correlated with a worse prognosis in terms of metastasis and survival [49]. High bacterial abundance was positively associated with immunosuppression and activation of oncogenic pathways. Importantly, RNA-sequencing showed significant differences in pancreatic microbiome abundance based on the smoking status and gender, two major risk factors of PDAC [49]. Riquelme et al. analyzed the effect of tumor microbiome diversity and composition on the survival of patients with pancreatic cancer [12]. The tumor microbiome of surgically resected patients with PDAC who survived more than 5 years after surgery (long-term survival, LTS) was compared to the stage-matched microbiome of patients who survived less than 5 years after surgery (short-term survival, STS). Taxonomic profiling via 16S rRNA gene sequencing revealed that alpha diversity of the tumor microbiome was significantly higher in the LTS patients compared with the STS cohort. This means individuals with high alpha diversity had significantly prolonged overall survival (median survival: 9.66 years) compared to those with low alpha diversity (median survival: 1.66 years). The tumor microbiome composition also differentiated cohorts of patients with short-term and long-term survival. The LTS patients showed enrichment on *Pseudoxanthomonas*, *Saccharopolyspora,* and *Streptomyces*. The combination of these three taxa + *Bacillus clausii* species resulted in an AUC 97.5, predicting long-term survivorship in PDAC patients [12]. These data suggest that the tumor microbiome alpha and beta diversity may serve as a potential predictor of survival outcome in PDAC patients undergoing surgical resection. Moreover, these results were confirmed in an animal ortothopic tumor model. A significant reduction in tumor growth in mice that received fecal microbiota transplantation from LTS donors was observed, compared with mice transplanted with stools from STS donors or healthy donors [12].

Analysis of the intraoperative bile culture of patients undergoing surgery due to adenocarcinoma of the head of the pancreas revealed that an increasing number of bacterial species found in bile was correlated with decreased progression-free survival (PFS) [41]. Importantly, the presence of *Klebsiella pneumonia* (*K. pneumonia*), a member of the class of *Gammaproteobacteria*, was detected in 35% of patients undergoing surgery. Compared to *K. pneumoniae*-negative patients, *K. pneumoniae*-positive patients had a larger tumor size (mean 29.9 mm vs. 23.7 mm; *p* = 0.003), reduced PFS (16.5 vs. 20.7 months; *p* = 0.032), and no improvement in PFS after gemcitabine therapy (13.2 vs. 19.5 months; *p* = 0.137).

A certain gut microbiome profile could be a predictor of the postoperative complication rate after pancreatic surgery. In a prospective pilot study of 32 patients, preoperative and postoperative gut microbiome analysis showed that an increase in *Akkermansia*, *Aeromonas*, *Enterobacteriaceae*, *Bacteroidales,* and a decrease in *Lachnospiraceae*, *Prevotella*, *Faecatitalea,* and *Bacteroides* at any time point were more frequently associated with a higher number of postoperative complications, a prolonged hospital stay, and a longer stay in the intensive care unit [42].

Reliable prognostic biomarkers in pancreatic cancer could modify the treatment method depending on the cancer biology. The data obtained so far strongly suggest the diversity and composition of the tumor microbiome may be useful as a prognostic tool for determining survival among PDAC patients. Moreover, these results imply that gut microbiome sequencing may be a very promising diagnostic tool in patients referred to surgery and modification of the microbiome before treatment could decrease the risk of complications related to pancreatic surgery. We definitely need more prospective multicenter studies to confirm these results.

## 7. The Usefulness of Microbiota Modification in the Therapy of Pancreatic Cancer

PDAC is refractory to most of the conventional therapeutic strategies [50]. Chemotherapy mainly involves 5-fluorouracil, oxaliplatin, and irinotecan (FOLFIRINOX) and gemcitabine monotherapy or a combination with nab-paclitaxel; however, the results of this treatment are not very impressive [51,52]. Bacterial species belonging to the *Gammaproteobacteria*, such as *K. pneumonia*, possess the enzyme cytidine deaminase, which metabolizes gemcitabine to an inactive form. Using the animal model of colon cancer, Geller et al. demonstrated that the *Gammaproteobacteria* present in the tumor was responsible for the resistance to gemcitabine, and this effect was abolished after the use of antibiotics [16]. Moreover, based on an analysis of 113 human PDAC samples obtained during surgery, they detected bacterial DNA in 76% of cancer cases and only 15% of cases in healthy pancreata. The most commonly identified species (51.7% of all readings) belonged to the *Gammaproteobacteria* class. This study confirmed that bacteria are a component of the pancreatic tumor microenvironment and may be involved in the resistance to chemotherapy. It has been shown that postoperative quinolone treatment in *K. pneumoniae*-positive PDAC patients resulted in improved overall survival and patients with quinolone-resistant *K. pneumoniae* had shorter PFS than those with quinolone-sensitive bacteria strains (9.1 vs. 18.8 months; *p* = 0.001) [41]. A retrospective analysis of patients with PDAC showed that antibiotics significantly improved the overall survival (13.3 vs. 9.0 months, *p* = 0.0001) and PFS (4.4 vs. 2 months, *p* < 0.0001) in individuals with metastatic disease, but not in patients who underwent tumor resection [53]. The effect was even more visible among patients who received concomitant gemcitabine-based chemotherapy. The observations that the response to a cancer drug can be increased by co-administration of antibiotics suggests that such combinations require additional clinical testing. On the other hand, there is a potential risk that ablation of the bacteria involved in gemcitabine metabolism may increase the effective dose of gemcitabine, which will contribute to the increased toxicity of the therapy. A re-analysis of the MPACT trial showed that in metastatic PDAC patients treated with gemcitabine, antibiotic exposure was associated with an increased risk of adverse events, mainly hematological (anemia, thrombocytopenia, leukopenia, neutropenia) and gastrointestinal [54]. More studies are needed to determine whether this observation merits any change in clinical practice.

Despite showing a beneficial effect in some cancers, immunotherapy has proved to be ineffective in PDAC, except in rare cases with a mismatch repair deficiency [55,56,57]. Programmed death receptor 1 (PD-1) is expressed in T-cells, B-cells, and macrophages. Its binding to PD-L1 ligands, which are found among others on cancer cells, inhibits the activation of the immune system, and promotes tumor progression. Checkpoint inhibitors are drugs that interfere with the function of the PD-1 receptor, preventing it from binding to the ligand. Currently, clinical trials targeting T cells or other checkpoint receptors were ineffective in PDAC [56]. Pushalkar et al. found that bacterial ablation with an antibiotic induced immunogenic reprogramming of the tumor microenvironment and upregulated PD-1 expression in intratumoral CD4+ and CD8+ T cells in mouse models [13]. Ablative oral antibiotics coupled with αPD-1 therapy were synergistically protective in a mice orthotopic cancer model. The combined therapy also resulted in enhanced intratumoral CD4+ and CD8+ T-cell activation. These data suggest that oral antibiotics in combination with checkpoint-directed immunotherapy may be an attractive strategy for experimental therapeutics in patients with PDAC. However, we must approach such conclusions with great caution. Antibiotics, apart from the undoubted benefits of their use, also have many negative effects on human health. The decreased number and diversity of gut and other niche microbiota caused by antibiotics is one of the factors leading to the development of serious diseases, including the *Clostridioides difficile* infection, obesity, inflammatory bowel disease, asthma, and cancers [58,59]. Moreover, it is estimated that 700,000 people die each year from drug-resistant infections, partially due to the overuse of broad-spectrum antibiotics [60]. The effect of antibiotics on cancer immunotherapy is also not fully understood. In a retrospective study of 291 patients with advanced tumors (melanoma, non-small cell lung cancer, renal cell carcinoma) treated with immune checkpoint inhibitors, antibiotics were shown to be an independent negative predictor of overall survival and SFP [61]. This effect has been greatly enhanced with the cumulative use of antibiotics and could be related to dysbiosis caused by antibiotics. Undoubtedly, further research is needed to analyze the impact and safety of antibiotics used to modify the microbiota in PDAC therapy.

## 8. Conclusions

Microbiota is an integral part of the PDAC environment and plays a pivotal role in cancer development and progression. Microbe-induced inflammation affects oncogenic signaling, tumor cell metabolism, and immune response in PDAC. Microbiome profiling has the potential to serve as a biomarker and identify individuals at risk for pancreatic cancer. Microbiome modulation plus immunotherapy is a novel approach; however, further research is needed in the future to confirm its usefulness in PDAC.

There are still many questions to be answered. We need to recognize specific microbiome components that are responsible for cancer development or maintenance and target them. What is the mechanism of action of these microbes—local or remote through systemic inflammation? How can microbiome-based screening tests be defined for PDAC? Finally, how should we deal with patients who have a PDAC-predisposing microbiome, but whose imaging tests show no evidence of disease? Charles Darwin stated “It is not the strongest of the species that survives, nor the most intelligent; it is the one most adoptable to change”. Microbiota have adapted to inhabit the pancreas and promote carcinogenesis. Our role is to reverse dysbiosis and fight against pancreatic cancer.

## Figures and Tables

**Table 1 jcm-10-05624-t001:** Common features of the microbiota differences and their consequences in patients with PDAC and precancerous lesions.

Anatomical Location of the Microbiota	Microbiota Differences in PDAC and Possible Clinical Consequences
Oral	
○ Higher microbial diversity in PDAC ○ Distinct composition from normal oral microbiota ○ Periodontal disease increases the risk of PDAC	Bacteroidetes [29,30] *Porphyromonas gingivalis*—↑ in PDAC Proteobacteria [30,40] *Aggregatibacter actinomycetemcomitans—*↑in PDAC [30] *Neisseria elongata*—↓ in PDAC [40] *Fusobacteria* [30] *Leptotrichia*—↓ in PDAC Firmicutes [43] *Streptococcus mitis*—↓ in PDAC Candida—↑ in PDAC [20]
Pancreas	
Intraductal papillary mucinous neoplasms (IPMNs) ○ Co-occurrence of oral bacterial species.	*Fusobacteria* [18] *Fusobacterium nucleatum*—in cyst fluid from IPMN with high-grade dysplasia. *Firmicutes* [18] *Granulicatella adiacensin*—in cyst fluid from IPMN with high-grade dysplasia.
PDAC ○ Distinct composition from normal pancreas ○ High diversity promotes long-term survival (LTS) ○ Low diversity promotes short-term survival (STS)	*Proteobacteria* [12,13,16] *Gammaproteobacteria*—contributes to gemcitabine resistance [16] *Pseudomonas* [13] *Elizabethkingia* [13] *Pseudoxanthomonas*—predicts LTS [12] *Bacteroidetes* [12,13] Bacteroidea—predicts STS [12] *Firmicutes* [12,13] *Bacillus clausii*—predicts LTS [12] *Clostridia*—predicts STS [12] *Actinobacteria* [12,13] *Streptomyces*—predicts LTS [12] *Saccharopolyspora*—predicts LTS [12] *Malassezia* [14] *Roseoloviruses*, *Lymphocryptoviruses*, *Gammaretroviruses* [21]
Gut	
○ Lower microbial diversity in PDAC ○ Distinct composition from normal gut microbiota ○ Reduced progression-free survival (PFS) ○ Higher rate of postoperative complications	*Proteobacteria* [13,41,42,43] *Klebsiella pneumoniae*—↑ tumour size, reduced PFS, resistance to gemcitabine therapy [41] *Aeromonas, Enterobacteriaceae*—↑rate of postoperative complications [42] Firmicutes—↓ in PDAC [43,44] *Fusobacteria* [13] *Verrucomicrobia* [13,44] *Akkermansia*—↑ rate of postoperative complications [42] *Actinobacteria* [13] *Bifidobacterium pseudolongum*—accelerated oncogenesis in Kras mice [13] *Bacteroidetes* [13,42,43] *Prevotella* [13] *Bacteroides* [13,42]—↑rate of postoperative complications *Hepatitis B and C virus* [25]
Blood	
○ Use of extracellular vesicles in PDAC	*Verrucomicrobia*, *Deferribacteres*, *Bacteroidetes* ↑ in PDAC [45] *Actinobacteria* ↓ in PDAC [45]

## Data Availability

No new data were created or analyzed in this study. Data sharing is not applicable to this article.

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
