# Peer review of "Microbiome—Friend or Foe of Pancreatic Cancer?"

_jcm, 2021, doi:10.3390/jcm10235624_

Round 1
Reviewer 1 Report
This review is focused on the microbiota role in the pancreas carcinogenesis. The authors describe different trials of mice and humans which are devoted to microbiome participation in PDAC pathogenesis. The PDAC is characterized by extremely poor prognosis and, nowadays, is the one of the crucial issues in the healthcare. Taken into account the growing evidence of microbiome participation in immunotherapy response for a number of cancers, the paper’s relevance is not in dispute.
There are several comments and minor suggestions below:
- In discussion it would be logical to suppose own critical author’s opinion about the limiting factor to incorporation “predictable gut microbiome composition” into clinical practice in the each of describing carcinogenesis steps.
- In discussion there is recommendation to present or perform analysis of common microbiota features from “high risk” pre-cancer patients, STS patients after surgery and after metastatic spread.
- There is recommendation to perform the common scheme for all parts of review.
- Abstract: «There is now increasing evidence that the human microbiome, which is involved in many physiological functions, including the regulation of metabolic processes and the modulation of the immune system, is strongly associated with pancreatic oncogenesis. » - Please, clarify the definition of the "strong association". In this point, it does it mean there are proven "microbiota oncоgenic drivers". Is it correct?
- Introduction: «However, any disturbances in microbiota composition, also known as dysbiosis, are associated with numerous diseases, such as obesity, diabetes, inflammatory bowel disease and cancers, like laryngeal, gastric, colorectal and liver cancer.»- In frame of cancer "association" is better to use participation. Otherwise, it has to be explained what the first event: microbiome stimulate cancer or oncogenesis changed microbiome.
- Part 4: «In an animal model of orthotopic pancreatic cancer, mice that 219 received fecal microbiota transplantation from long-term survival patients had significantly higher numbers of CD8+ T cells, activated T cells (CD8+IFNg+ T cells) within the 221 tumor, as well as higher serum levels of interferon-g and IL-2 compared to mice that received fecal microbiota from short-term survival patients» - Are there the lifespan of these miсe?
- Part 6: «Importantly, the presence of Klebsiella pneumonia (K. pneumonia), a member of the class of 338 Gammaproteobacteria, was detected in 35% of patients undergoing surgery. K. pneumoniae 339 colonization was associated with a larger tumor size, reduced PFS, and a reduced effect 340 of adjuvant gemcitabine on PFS.» - For adjuvant settings it is recommended to use DFS or RFS. In addition, please, add specific values with range and statistical significance for “PFS”, clarify the “effect” definition.
- Part 7: «This study confirmed that bacteria are a component of the pancreatic tumor 364 microenvironment and can play a key role in resistance to chemotherapy. It has been 365 shown that postoperative quinolone treatment in K. pneumoniae positive PDAC patients 366 resulted in improved overall survival and patients with quinolone-resistant K. pneumoniae 367 had shorter PFS than those with quinolone-sensitive bacteria strains (9.1 vs. 18.8 months; p=0.001) ». – It has been recommended to change the wording “play a key role in resistance” because of 9 months clinical benefit the likelihood one the factors which influence on therapy efficiency but is hard to say that is the key.
Conclusion: «Microbiome modulation plus immunotherapy is a novel approach to alter the grim 417 prognosis of PDAC.» - The assertion does not follow from the text above and should be changed/deleted or to be described in the review.
Author Response
Reviewer’s comment #1: This review is focused on the microbiota role in the pancreas carcinogenesis. The authors describe different trials of mice and humans which are devoted to microbiome participation in PDAC pathogenesis. The PDAC is characterized by extremely poor prognosis and, nowadays, is the one of the crucial issues in the healthcare. Taken into account the growing evidence of microbiome participation in immunotherapy response for a number of cancers, the paper’s relevance is not in dispute.
Answer: We thank you for this positive and encouraging comment.
Reviewer’s comment #2: In discussion it would be logical to suppose own critical author’s opinion about the limiting factor to incorporation “predictable gut microbiome composition” into clinical practice in the each of describing carcinogenesis steps.
Answer: We greatly appreciate this suggestion. We have added our critical opinion on the limiting factors to incorporate “predictable gut, pancreatic and oral microbiome into clinical practice” to every section of the manuscript (line 123; 159; 241; 336; 393), except for section # 7 – “The usefulness of microbiota modification in the treatment of pancreatic cancer”, as this section already contained our opinion.
Reviewer’s comment #3: In discussion there is recommendation to present or perform analysis of common microbiota features from “high risk” pre-cancer patients, STS patients after surgery and after metastatic spread.
Answer: We have attached “Table 1” (line 258) to our manuscript that lists the most common microbiome changes in the oral cavity, gut, pancreas and blood in patients with pancreatic cancer and precancerous lesions. Where available, we also added information regarding the impact of microbiome changes on treatment outcomes and the incidence of post-treatment complications.
Reviewer’s comment #4: There is recommendation to perform the common scheme for all parts of review.
Answer: To explain and clarify the topic of this review, we have added a short description of what the individual sections of the manuscript will cover at the end of the introduction section (Line 52). Furthermore, we have added our individual comments at the end of most sections to unify the individual parts of the manuscript (line 123, 159; 241; 336; 393).
Reviewer’s comment #5: Abstract: «There is now increasing evidence that the human microbiome, which is involved in many physiological functions, including the regulation of metabolic processes and the modulation of the immune system, is strongly associated with pancreatic oncogenesis.» - Please, clarify the definition of the "strong association". In this point, it does it mean there are proven "microbiota oncоgenic drivers". Is it correct?
Answer: Thank you for this comment. In our manuscript, we have reviewed numerous studies, both experimental and clinical, describing the relationship between the microbiome and pancreatic cancer. However, there is no conclusive evidence at the moment that confirms the direct involvement of the microbiome in oncogenesis. Therefore, we changed the content of the sentence. «There is now increasing evidence that the human microbiome, which is involved in many physiological functions, including the regulation of metabolic processes and the modulation of the immune system, is possibly linked to pancreatic oncogenesis. » (line 11)
Reviewer’s comment #6: Introduction: «However, any disturbances in microbiota composition, also known as dysbiosis, are associated with numerous diseases, such as obesity, diabetes, inflammatory bowel disease and cancers, like laryngeal, gastric, colorectal and liver cancer.»- In frame of cancer "association" is better to use participation. Otherwise, it has to be explained what the first event: microbiome stimulate cancer or oncogenesis changed microbiome.
Answer: The content of the sentence has been changed as suggested by the reviewer. «However, any disturbances in microbiota composition, also known as dysbiosis, are associated with numerous diseases, such as obesity, diabetes, and inflammatory bowel disease. Moreover, the participation of dysbiosis in cancers, like laryngeal, gastric, colorectal, and liver cancer has recently been emphasized.» (line 32)
Reviewer’s comment #7: Part 4: «In an animal model of orthotopic pancreatic cancer, mice that 219 received fecal microbiota transplantation from long-term survival patients had significantly higher numbers of CD8+ T cells, activated T cells (CD8+IFNg+ T cells) within the 221 tumor, as well as higher serum levels of interferon-g and IL-2 compared to mice that received fecal microbiota from short-term survival patients» - Are there the lifespan of these miсe?
Answer: In the article by Riquelme et al., the research protocol (administration of antibiotics to mice, fecal microbiota transplantation, orthotopic implantation of tumor cells) was established for 2 months. After this time, all mice were euthanized and the obtained organs were analyzed. The effect of fecal microbiota transplantation on the lifespan of mice has not been studied.
Reviewer’s comment #8: Part 6: «Importantly, the presence of Klebsiella pneumonia (K. pneumonia), a member of the class of 338 Gammaproteobacteria, was detected in 35% of patients undergoing surgery. K. pneumoniae 339 colonization was associated with a larger tumor size, reduced PFS, and a reduced effect 340 of adjuvant gemcitabine on PFS.» - For adjuvant settings it is recommended to use DFS or RFS. In addition, please, add specific values with range and statistical significance for “PFS”, clarify the “effect” definition.
Answer: We agree with the reviewer that DFS or RFS is a better parameter for adjuvant treatment. However, in the cited study, Weniger et al. only used PFS with respect to gemcitabine treatment. We also added numerical values, statistical significance and replaced the word "effect" with "no improvement" to make the text more transparent (line382).
Reviewer’s comment #9: Part 7: «This study confirmed that bacteria are a component of the pancreatic tumor 364 microenvironment and can play a key role in resistance to chemotherapy. It has been 365 shown that postoperative quinolone treatment in K. pneumoniae positive PDAC patients 366 resulted in improved overall survival and patients with quinolone-resistant K. pneumoniae 367 had shorter PFS than those with quinolone-sensitive bacteria strains (9.1 vs. 18.8 months; p=0.001) ». – It has been recommended to change the wording “play a key role in resistance” because of 9 months clinical benefit the likelihood one the factors which influence on therapy efficiency but is hard to say that is the key.
Answer: We greatly appreciate this suggestion. We agree with the reviewer's opinion that the current state of our knowledge regarding the influence of microbiota on chemotherapy does not allow us to make such conclusions. Therefore, we have replaced "can play a key role in resistance" with the phrase "may be involved in the resistance" (line 413).
Reviewer’s comment #10: Conclusion: «Microbiome modulation plus immunotherapy is a novel approach to alter the grim 417 prognosis of PDAC.» - The assertion does not follow from the text above and should be changed/deleted or to be described in the review.
Answer: We agree with the reviewer that the current results of immunotherapy with microbiome modulation are inconclusive and do not allow their routine use in PDAC. Therefore, we changed the sentence as suggested by the reviewer (line 466).
Reviewer 2 Report
Daniluk et al. reported relationship between microbiome and pancreatic cancer. This manuscript is well reviewed, but there are some reviews of the relationship in 2021.1-3) There are little new findings in this manuscript.
- Page 4, line 178. After “periodontal disease”, abbereviation should be added.
- Page 8, line 397. Is “PDA” “PDAC”?
- Basu M, Philipp LM, Baines JF, Sebens S. The Microbiome Tumor Axis: How the Microbiome Could Contribute to Clonal Heterogeneity and Disease Outcome in Pancreatic Cancer. Front Oncol. 2021;11:740606. doi: 10.3389/fonc.2021.740606.
- Li JJ, Zhu M, Kashyap PC, Chia N, Tran NH, McWilliams RR, Bekaii-Saab TS, Ma WW. The role of microbiome in pancreatic cancer. Cancer Metastasis Rev. 2021;40:777-789. doi: 10.1007/s10555-021-09982-2.
- Abdul Rahman R, Lamarca A, Hubner RA, Valle JW, McNamara MG. The Microbiome as a Potential Target for Therapeutic Manipulation in Pancreatic Cancer. Cancers (Basel). 2021;13:3779. doi: 10.3390/cancers13153779.
Author Response
Reviewer’s comment #1: Daniluk et al. reported relationship between microbiome and pancreatic cancer. This manuscript is well reviewed, but there are some reviews of the relationship in 2021.1-3) There are little new findings in this manuscript.
- Basu M, Philipp LM, Baines JF, Sebens S. The Microbiome Tumor Axis: How the Microbiome Could Contribute to Clonal Heterogeneity and Disease Outcome in Pancreatic Cancer. Front Oncol. 2021;11:740606. doi: 10.3389/fonc.2021.740606.
- Li JJ, Zhu M, Kashyap PC, Chia N, Tran NH, McWilliams RR, Bekaii-Saab TS, Ma WW. The role of microbiome in pancreatic cancer. Cancer Metastasis Rev. 2021;40:777-789. doi: 10.1007/s10555-021-09982-2.
- Abdul Rahman R, Lamarca A, Hubner RA, Valle JW, McNamara MG. The Microbiome as a Potential Target for Therapeutic Manipulation in Pancreatic Cancer. Cancers (Basel). 2021;13:3779. doi: 10.3390/cancers13153779.
Answer: Thank you very much for this insightful comment. Due to the lack of significant improvement in the diagnosis and therapy of PDAC, the problem of mutual interactions between the microbiome and pancreatic oncogenesis has gained enormous popularity recently, which is reflected in the number of publications on this topic. According to the PubMed website, the number of publications on this subject has doubled this year in comparison to 2018. The articles cited by the reviewer are undoubtedly very valuable and analyze in detail the role of microbiota in the pathogenesis, diagnosis and treatment of PDAC. We are aware of the limitations of our work, but firmly believe that our study can provide additional information on the role of the microbiome in PDAC. The article by Basu et al. focused mainly on the influence of the microbiome on the Epithelial-Mesenchymal-Transition, the abundance of Cancer Stem Cells and on the high intratumor heterogeneity. Compared to the articles by Li et al. and Rahman et al., our manuscript contains some additional information, such as potential mechanisms leading to microbiome colonization of the pancreas or the effect of virus colonization in pancreatic tumors. We additionally discussed new methods for the early detection of pancreatic cancer, for example the assessment of extracellular vesicles from blood samples and utility pancreatic formalin-fixed paraffin-embedded (FFPE) samples obtained during endoscopic ultrasound fine needle biopsy (EUS-FNB) to estimate the pancreatic microbiome. Based on our experience in the animal model of PDAC, we also presented the potential impact of chronic inflammation on cancer development. Taking into account the arguments presented by us, we believe that our study fills the gaps that have not been addressed in the cited works and may constitute a significant supplement to the knowledge on the influence of the microbiome on the development of PDAC.
Reviewer’s comment #2: Page 4, line 178. After “periodontal disease”, abbreviation should be added.
Answer: The authors added the appropriate abbreviation (line194)
Reviewer’s comment #3: Page 8, line 397. Is “PDA” “PDAC”?
Answer: We apologize for the misunderstanding. Of course, our intention was to write the abbreviation for PDAC. We have made the appropriate correction in the text (line 446).
Round 2
Reviewer 2 Report
This report has been revised well according to reviewers’ comments.
This manuscript is a resubmission of an earlier submission. The following is a list of the peer review reports and author responses from that submission.
Round 1
Reviewer 1 Report
MAJOR CRITIQUES
- Line 54: This section needs to include the clinical findings of Riquelme et al., Cell 2019 as well.
- Line 158: Introducing P. gingivalis at this point seems out of place. The readers need to know beforehand why this oral microbe is relevant for pancreatic cancer, perhaps it may be helpful to mention the clinical studies at this point rather than in the biomarker section. Additionally, instead of citing another review article, authors should cite the original research articles that demonstrated the role of TLR4 and P. gingivalis.
- Line 197: Authors should add potential of blood microbial signatures as biomarkers- Poore, G.D. et al. Nature 579, 567–574 (2020) doi.org/10.1038/s41586-020-2095-1
- Line 343: Systemic use of antibiotics can give arise to several undesired effects including rise of resistant bacteria, gastrointestinal complications, etc. Authors should discuss the advantages and disadvantages of global targeting of the microbiome with systemic delivery of broad-spectrum antibiotics
MINOR CRITIQUES
- Line 91: For the virome section authors may consider discussing the following paper: Zapatka, M. et al. Nat Genet 52, 320–330 (2020) doi.org/10.1038/s41588-019-0558-9
- Line 115: Authors describe two studies with opposing results but do not discuss possible mechanisms or explanations for the discrepancies.
- Line 132: Needs citation
- Line 153: Systemic antibiotic treatment will deplete not only the gut microbes, but at other sites as well
- Line 249: In the gut microbiome biomarker section, authors should also include the findings from Half, E. et al. Sci Rep 9, 16801 (2019) doi.org/10.1038/s41598-019-53041-4
- Line 310: Authors may add the following study to this section- Mohindroo C. et al. Cancer Med. 2021;10(15):5041-5050. doi:10.1002/cam4.3870